# Modelling of Phase Contrast Imaging with X-ray Wavefront Sensor and Partial Coherence Beams

**DOI:** 10.3390/s20226469

**Published:** 2020-11-12

**Authors:** Ginevra Begani Provinciali, Alessia Cedola, Ombeline de La Rochefoucauld, Philippe Zeitoun

**Affiliations:** 1LOA, ENSTA Paris, CNRS, Ecole Polytechnique IP Paris, 828 Boulevard des Maréchaux, 91120 Palaiseau, France; 2Institute of Nanotechnology-CNR c/o Physics Department, Sapienza University of Rome, Piazzale Aldo Moro 5, 00185 Rome, Italy; alessia.cedola@cnr.it; 3Imagine Optic, rue François Mitterrand, 33400 Talence, France; odlrochefoucauld@imagine-optic.com

**Keywords:** phase-contrast imaging, Hartmann sensor, simulation, partial coherence

## Abstract

The Hartmann wavefront sensor is able to measure, separately and in absolute, the real δ and imaginary part β of the X-ray refractive index. While combined with tomographic setup, the Hartman sensor opens many interesting opportunities behind the direct measurement of the material density. In order to handle the different ways of using an X-ray wavefront sensor in imaging, we developed a 3D wave propagation model based on Fresnel propagator. The model can manage any degree of spatial coherence of the source, thus enabling us to model experiments accurately using tabletop, synchrotron or X-ray free-electron lasers. Beam divergence is described in a physical manner consistent with the spatial coherence. Since the Hartmann sensor can detect phase and absorption variation with high sensitivity, a precise simulation tool is thus needed to optimize the experimental parameters. Examples are displayed.

## 1. Introduction

In recent years, the evolution of X-ray imaging technologies allowed for in-vivo/ex-vivo visualization of biological tissues opened new prospective and innovative applications [1,2,3]. In conventional X-ray imaging techniques, the rays are attenuated by materials with different densities and chemical compositions according to Beer Lambert’s law [4]. The image contrast originates from the differential absorption properties of the sample. In classic 2D radiography, the intensity at any point of the image is given by the integration of the absorption of rays propagating throughout the whole sample. Therefore, in radiography, the image is the superposition of each element inside the sample perpendicular to the beam propagation.

X-ray tomography overcomes this limitation providing 3D images that can be virtually sliced in all the directions. In tomography, a series of 2D images of the sample illuminated by an X-ray beam are collected for a large number of angles of view. A reconstruction algorithm is then used starting from this stack of 2D images to obtain a volumetric visualization of the sample. Thus, X-ray tomography is a non-destructive, high-resolution technique that provides a tridimensional reconstruction of the object.

X-ray imaging of weakly absorbing material (like biological tissues) is an open challenge in imaging, since absorption contrast is very low and often exhibits weak variations all over the sample. This makes very difficult to differentiate the different elements contained inside the sample. Image contrast can be increased up to 3 orders of magnitude, taking advantage of the phase modulation induced by the sample on a coherent or partially coherent X-ray beam. In the hard X-ray region, the phase contrast is predominant over the absorption contrast [5], potentially providing detailed images of the sample. Moreover, since light materials are almost transparent to X-rays, the dose delivered to the sample is consistently lower with respect to absorption-based X-ray imaging. Tomography can be additionally combined with phase imaging allowing for a full spatial coverage at different depths inside the investigated specimen. Many techniques have been implemented to perform X-ray Phase Contrast Tomography (XPCT) like free-space propagation set-ups [6,7], coded aperture [2], Talbot and Talbot-Laue [8,9], and interferometry [10].

The majority of phase contrast methods provide a qualitative analysis of the sample. The principal limitation concerning free-space propagation phase contrast imaging is that the outcome is proportional to the ratio δ/β of the real (δ) and imaginary part (β) of the refraction index. This implies that it is not possible to separate the contribution of δ and β with only one image, like phase and absorption of the object. Losing this information prevents the calculation of the exact density of the material or retrieving its chemical composition.

We present a novel, high-sensitivity X-ray quantitative phase contrast imaging system based on a Hartmann wavefront sensor (see [11], this issue). The Hartmann sensor is a device that is commonly used for wavefront sensing [12]. It is able to independently reconstruct the phase and the amplitude of the beam with high accuracy. In the Hartmann sensor, the wavefront is sampled by a grid of regularly spaced holes that generates an array of beamlets that are intercepted by a 2D detector, leading to an array of spots. A sample located in the path creates a local change of the beamlets propagation angle (i.e., wave vector k), shifting the spots on the detector with respect to their initial positions. Measuring the local displacements (∆x, ∆y) of the spots allows the retrieval of the local wave vectors (k_x_ and k_y_). Integration of the k_x_ and k_y_ maps produces the wavefront map. The sample absorption can also be calculated by integrating the signal correspondent to each hole and dividing it by the signal recorded without the sample. Therefore, a wavefront sensor generates the absorption and the phase maps of the sample. The system works on a single acquisition, making the measurement procedure fast and stable. Since the system is lensless, it may work with high sensitivity over a wide range of energies as well as with monochromatic or polychromatic beams. Lastly, it can be easily coupled with tomographic setups by replacing the 2D detector by the wavefront sensor to obtain 3D images of both the phase and the amplitude.

Many structural and experimental parameters need to be optimized to design a Hartmann wavefront sensor dedicated to imaging applications. In particular, the architecture of the Hartman hole plate, the distances between the different elements of the set-up and the source properties are some of the crucial issues that determine the performances of the entire system.

Here we present a simulation code that reproduces the outcome of a real set-up. To test the versatility of the Hartmann sensor it is fundamental to accurately simulate the proprieties of different kind of sources, paying particular attention on the definition of the degree of spatial coherence. The goal of this paper is to study the impact of coherence degree on the performances of EUV or X-ray Hartmann wavefront sensors. The general framework of optical coherence theory and related propagation of partially coherent beam is already well established [13,14,15]. We thus adjusted existing techniques to model the specific cases of Hartmann sensors combined with existing X-ray sources.

Many techniques are currently available to describe partially coherent beams by a superposition of coherent but mutually uncorrelated light beams [15,16,17]. In the Gaussian Schell model (GSM) [18] the beam represents a broad class of partially coherent beams, whose intensity and degree of coherence both satisfy Gaussian distributions. In the twisted Gaussian Schell model (TGSM) [19] partially coherent beams can carry two special kinds of phase (i.e., twist phase and vortex phase) besides the usual quadratic phase. Recently, a great deal of attention has been paid to partially coherent beams whose degree of coherence does not satisfy a Gaussian distribution, so-called partially coherent beams with nonconventional correlation functions [20]. An alternative approach consists of the representation of the wavefront to be propagated by a linear superposition of Gaussian beam wavefronts [21]. Each beam is propagated independently throughout the optical system and then superposed to calculate the propagated field.

We numerically studied an application of this technique for the imaging of weakly absorbing materials. A modified version of the superposition of Gaussians approach is described here. Before the propagation, each Gaussian beam is shifted to a random position chosen inside a larger Gaussian distribution (source size) and multiplied by a random phase factor. Then the propagation is applied and the electric fields added together at the end of each cycle to generate the final field. This article will describe the implementation of a partially coherent source using Fresnel propagators able to deal with any sources from fully coherent to the incoherent. Results from the transmission radiography of a test object are shown. Discussions about different designs of Hartmann sensors are thus performed.

## 2. Materials and Methods

### 2.1. Fresnel Propagator

The radiation emitted from the source was propagated step-by-step to each optical element using a Fresnel Propagator. The wave field *U(x*,*y*) was calculated from the diffraction of a complex aperture placed in the (*ε*,*η*) plane that was illuminated by the field *U*(*ε*,*η*) in the positive z direction (Figure 1). The propagator was implemented following its classical derivation from the Huygens-Fresnel principle [22]:(1)$$U\left(x,y\right)=\frac{z}{j\lambda }\overset{+\infty }{\underset{-\infty }{\iint }}U\left(\varepsilon ,\eta \right)\frac{e^{jkr}}{r^{2}}d\varepsilon d\eta$$
where *λ* is the wavelength, *k* is the wave vector, *U*(*x*,*y*) and *U*(*ε*,*η*) are the wave fields in the (*x*,*y*) and in the (*ε*,*η*) plane, respectively, *z* is the propagation distance and where the distance r is given by:(2)$$r=\sqrt{z^{2}+{\left(x-\varepsilon \right)}^{2}+{\left(y-\eta \right)}^{2}}$$

To reduce the Huygens-Fresnel to a simpler expression, we introduce a binomial expansion for the distance r and keep only the constant and quadratic terms of the expansion:(3)$$r\approx z\left[1+\frac{1}{2}{\left(\frac{x-\varepsilon }{z}\right)}^{2}+\frac{1}{2}{\left(\frac{y-\eta }{z}\right)}^{2}\right]$$

Inserting Equation (3) inside Equation (1) we can rewrite the field at (*x*,*y*) as a convolution in the following form:(4)$$U\left(x,y\right)=\underset{\left(-\infty ,+\infty \right)}{\iint }U\left(\varepsilon ,\eta \right)h\left(x-\varepsilon ,y-\eta \right)d\varepsilon d\eta$$
where the convolution kernel $h\left(x-\varepsilon ,y-\eta \right)$ is defined as:(5)$$h\left(x,y\right)=\frac{e^{jkz}}{j\lambda z}e^{\frac{jk}{2z}\left(x^{2}+y^{2}\right)}$$

Factorizing the term $e^{\frac{jk}{2z}\left(x^{2}+y^{2}\right)}$ outside the integral we find:(6)$$U\left(x,y\right)=\frac{e^{jkz}}{j\lambda z}e^{\frac{jk}{2z}\left(x^{2}+y^{2}\right)}\underset{\left(-\infty ,+\infty \right)}{\iint }\left\{U\left(\varepsilon ,\eta \right)e^{\frac{jk}{2z}\left(\varepsilon^{2}+\eta^{2}\right)}\right\}e^{-\frac{j2\pi }{\lambda z}\left(x\varepsilon +y\eta \right)}d\varepsilon d\eta$$

Equation (6) is the Fourier transform of the product of the initial complex field *U*(*ε*,*η*) and a quadratic phase factor $e^{\frac{jk}{2z}\left(\varepsilon^{2}+\eta^{2}\right)}$, constant terms are factorized outside the integral.

Equation (6) goes under the name of Fresnel diffraction integral. This approximation is valid when the observation distance r is many wavelengths larger than the aperture (r ≫ λ) and is called the near-field of Fresnel diffraction region.

The Fresnel propagator of the initial field after a distance z (*U_2_*(*x*,*y*)) is the inverse Fourier transform of the multiplication of the Fourier transform of the initial field *U*_1_(*x*,*y*) and the propagator$\;H\left(f_{x},f_{y}\right)$:(7)$$U_{2}\left(x,y\right)=FT^{-1}\left\{FT\left(U_{1}\left(x,y\right)*H\left(f_{x},f_{y}\right)\right)\right\}$$
where the direct and inverse Fourier transform are indicated with *FT* and *FT*^−1^, respectively, *U*_1_(*x*,*y*) is the initial field and the propagator $H\left(f_{x},f_{y}\right)$ can be expressed by:(8)$$H\left(f_{x},f_{y}\right)=e^{-j\pi \lambda z\left(f^{2}{}_{x}+f^{2}{}_{y}\right)}e^{jkz}$$
where $\left(f_{x},f_{y}\right)$ are the spatial frequencies and *z* is the propagation distance.

In this way, we are able to compute the convolution integral Equation (6) as series of simple Fourier transforms that make the code faster.

### 2.2. Propagator Validation

The first step to validate the implementation of our Fresnel propagator is to simulate the propagation of a fully coherent beam using an object leading to well-known diffraction patterns. We looked at the diffraction pattern of a coherent source from a rectangular aperture for various normalized distances from the aperture, as represented by different Fresnel numbers. The Fresnel number *N_f_* = *a^2^/zλ* is the ratio between the aperture size (*a*) and the product of the distance of the screen from the aperture (*z*) and the incident wavelength (*λ*). The simulation was performed with *λ* = 1.38 × 10^−10^ m, *M* = 1000 × 1000 points, the distance *z* and the aperture size were changed for each *N_f_*. Diffraction patterns generated by our code are reported in Figure 2a while results tabulated in [22] (p. 86) are given in Figure 2b for *N_f_* = 1, 4, 10. For each plot the size of the aperture is given.

### 2.3. Source Coherence

The source is composed of a series of Gaussian sources with standard deviation *σ*. The beams are propagated independently up to the detector plane, where the electric field of every beam is summed together. Prior to propagation, each beam is multiplied by a random phase factor *φ*(*x*,*y*) and is shifted in a random position chosen inside a larger Gaussian distribution with standard deviation *σ_B_* that defines the source size. Let us define N as the number of Gaussian beams inside the source, the final electric field expression being
(9)$$E^{tot}=\overset{N}{\underset{i=1}{\sum }}E(x_{i}^{r_{i}},y_{i}^{r_{i}})e^{i\phi_{i}^{r_{i}}(x_{i}^{r_{i}},y_{i}^{r_{i}})}$$
where $E^{tot}$ is total electric field,$\;E(x_{i}^{r_{i}},y_{i}^{r_{i}})$ is the electric field calculated in each position ($x_{i}^{r_{i}},y_{i}^{r_{i}}$) randomly distributed inside the source and $e^{i\phi_{i}^{r_{i}}\left(x_{i}^{r_{i}},y_{i}^{r_{i}}\right)}$ is a random phase factor.

We first simulated a fully coherent source (*N* = 1) and we observed its capability of creating a diffraction pattern after passing through a sharp edge (black curve, Figure 3a) looking at its normalized intensity pattern. The simulation was performed with the following parameters: wavelength *λ* = 1.38 × 10^−10^ m, number of points *M* = 3000, and a propagation distance *z* = 40 cm. The oscillations that can be seen on the black line of Figure 3a for a low number of Gaussians inside the source (*N* = 5) are due to its high level of coherence. The measured fringe visibility is related in fact to the magnitude of the complex degree of coherence within the illumination beam [23]. Taking the same experimental condition but increasing the number of Gaussian sources (*N* = 100), making the beam partially coherent, the oscillations vanished (blue line in Figure 3a) since they are averaged between many shifted beams. The diffraction pattern created by the object in the case of a coherent illumination corresponds to the one tabulated in the literature (Figure 3b) [24] (p. 180).

This example confirms that increasing *N* decreases the degree of spatial coherence of the source.

These two first examples prove the reliability of our numerical model to treat the Fresnel propagator in the case of a fully coherent or partially coherent illumination.

### 2.4. Computational Details

The simulation was performed using an Intel Xeon CPU E5-1650 v4@3.60GHz on a Windows environment with 64 Gb of RM installed. The average time for the simulation of each single beam is about 4 s, and in the case of 100 sources the simulation time was about 6.5 min. The code was edited in Python (version 3.5.2). The code was parallelized on 6 cores to obtain faster results.

## 3. Results and Discussion

### 3.1. Diffraction from a Test Object

To simulate a reasonable experimental situation, a simple object was taken as an example. The simulated sample was a 3D cylinder composed of Polymethylmethacrylate (PMMA). The material is described by its index of refraction $n\left(\omega \right)=1-\delta \left(\omega \right)-i\beta \left(\omega \right)$, where ω is the wave pulsation. The interaction of the incident beam with an object is described trough its transfer function. The wave electric field (*E*(*x*,*y*)) after the interaction with the object can be expressed as:(10)$$E\left(x,y\right)=A_{0}\left(x,y\right)e^{\frac{2\pi i\delta \Delta L}{\lambda }}e^{-\frac{2\pi \beta \Delta L}{\lambda }}$$
where ΔL is the object thickness, $A_{0}\left(x,y\right)$ is the amplitude of the incident wave, *λ* is the wavelength, δ and β are the real and imaginary parts of the index of refraction.

For the sake of simplicity, the thickness of the object is evaluated at each (*x*,*y*) point. The propagation inside the sample is not yet considered. Since the deflexion angles are very small, in the 100 nrad to µrad range, the deformation of the wavefront just at the end of the object remains negligible.

The simulation is performed at an incident energy of 9 keV, with *δ* = 3.29 × 10^−6^ and *β* = 5.69 × 10^−9^. The set-up is composed of a source composed of different numbers of initial Gaussian beams, a PMMA cylinder and a detector. The simulated plane is composed of 5000 × 5000 points and each Gaussian source has a standard deviation *σ* = 0.03 µm. The distance source-object is *z**_1_* = 5 cm and the distance object-detector is *z_2_* = 5 cm. The PMMA cylinder diameter is 50 µm and the pixel size of the detector is 0.13 µm. Figure 4 shows the diffraction pattern from a PMMA cylinder in the case of *N* = 1, 10, 100 and 1000 Gaussian sources. The general shape of the cylinder is well reproduced with a stronger absorption at its middle. For a low number of Gaussians, diffraction fringes are clearly visible outside the cylinder (Figure 4a). However, for higher *N*, in particular *N* = 1000, the beam outside the cylinder appears quite homogeneous (Figure 4d).

Horizontal plots (Figure 5) are shown to appreciate the evolution of the diffraction patterns between the different cases. Many contrasted oscillations can be seen in the case of fully coherent illumination (green line*, N* = 1), while with a small number of sources (red line, *N* = 10) the signal appears noisy due to the multiple interferences. An incoherent illumination can be simulated using a high number of sources (black line, *N* = 1000). In this case, the diffraction pattern is completely smoothed out due to the random superposition of the electric fields coming from the very large number of sources. The case *N* = 100 still exhibits very weak oscillations. The shape of the incident Gaussian beam can be seen at the centre of the intensity plot (Figure 5) since the low absorption properties of PMMA are not completely attenuating the incident beam.

The simulation results show that free-propagation imaging is simple for the extreme cases of completely coherent (Figure 5, green line) and incoherent illumination (Figure 5, black line). However, it becomes challenging when the source is partially coherent (Figure 5, red and blue lines).

### 3.2. Imaging with Very Compact Hartmann Sensor

The previous part validates the relationship between the coherence of the source and its effect on imaging procedure. This is an important finding in the understanding of the behavior of optical systems with respect to the characteristics of the illuminating source. The design of a highly sensitive wavefront sensor, such as the Hartmann sensor, requires the optimization of many parameters. The Hartmann mask is an absorbing plate with regularly spaced holes having 100% transmission (Figure 6). The reported images display beam after the propagation through a Hartmann mask with 3 µm square holes and 5 µm pitches. The simulated plane is composed of 5000 × 5000 points and each Gaussian source has standard deviation *σ* = 0.03 µm. The incident energy is set at 9 keV. The distance source-mask is *z*_1_ = 5 cm and the distance mask-detector is *z*_2_ = 1 cm.

The pixel size of the detector is 0.066 µm. Since the reconstruction of the wavefront is based on the integration of the signal of each beam to calculate their displacement [12,25], we are going to investigate the occurrence of diffraction and the cross-talk between adjacent holes when changing the coherence of the illuminating beam.

In Figure 7, images of the Hartmann patterns in the case of *N* = 1, 10, 100 and 1000 are reported. Diffraction effect from the hole edges is clear in the case of coherent illumination (*N* = 1, Figure 7a), while the displacement of the sub-sources associated with the interference of the propagated beams induces a strong change on the pattern *N* = 10 (Figure 7b). Decreasing the degree of coherence of the incident beam decreases the visibility of the diffraction pattern (*N* = 100, Figure 7c), and in the case of strongly incoherent illumination (*N* = 1000, Figure 7d) a Gaussian shape for the diffraction pattern of each hole can be seen. Such Hartmann patterns are characteristic of very compact wavefront sensors. They have been observed on the high numerical aperture EUV sensor [26,27].

Single line plots of each case shown in Figure 7 are reported in Figure 8. The mask parameters, like the spacing between two holes or their dimension, can be optimized with respect to the degree of coherence of the source. In fact, the efficiency of the wavefront reconstruction algorithms is strongly related to the spot shape. In the case of incoherent illumination (Figure 7d), approximating each spot with a Gaussian distribution will allow a fast and reliable reconstruction of the beam local deflection.

### 3.3. Imaging with Standard Hard X-ray Hartmann Sensors

In the Hartmann mask design studied in Section 3.2 the pitch is comparable with the aperture size and the detector was placed at 1 cm from the mask. Standard experimental conditions for Hartmann imaging usually requires a larger pitch and to move the dector plane further [23]. To meet these criteria, a second Hartmann mask was modelled with 3 µm square holes and 25 µm pitches. The simulated plane is composed of 5000 × 5000 points and each Gaussian source has standard deviation *σ* = 0.03 µm. The incident energy is set at 9 keV. The distance source-mask is *z*_1_ = 20 cm and the distance mask-detector is *z*_2_ = 20 cm. The detector pixel size is 0.13 µm. The results are displayed in Figure 9.

This configuration differs from the previous one in the sense that the detector is placed after the Fraunhofer distance, given by *L = 2 d*^2^*/λ*, where d is the aperture size. Within the modeling conditions, *L* = 0.13 m that is about half the distance mask-detector. As a consequence, the diffraction patterns for all the cases are exhibiting the well-known Airy pattern. It is still important to remember that for partially coherent sources, the pattern is convolved with the image of the source formed by each hole. For totally incoherent, the pattern is like in Figure 8 the image of the source. For cases of the highest degree of coherence, the diffraction of one hole extend far up to the neighboring hole induces deleterious cross-talk. For real wavefront sensors, the square holes are tilted at 22.5° to prevent the cross-talk [26,27,28]. In the full coherent case (Figure 9a) small oscillations can be seen for each spot, while they tend to disappear with incoherent illumination (Figure 9d). Such oscillations are not detectable on a standard Hartman sensor since the pixel size is about 1 µm, compared to 0.13 µm in the current modeling.

### 3.4. Talbot Effect

One of the well-known phenomena connected with coherent illumination of regularly spaced diffractive objects is the Talbot effect [24]. In 1881, Lord Rayleigh was the first to prove that this phenomenon is a consequence of the diffraction interference of highly spatially coherent (plane) waves after the interaction with a periodic structure, such as a grating. When a plane coherent wave interacts with a periodic structure, it reproduces the exact pattern of the grating after a certain distance, called Talbot distance. At the Talbot distance *z_T_*, all the diffraction orders are reinforced:(11)$$z_{T}=2m\frac{d^{2}}{\lambda }$$
where *d* is the period of the grating and *m* is a positive integer. Each value of m goes to a plane where the image of the periodic structure is reproduced. We perform a simulation of the Hartmann mask at the first Talbot plane with different degree of coherence (Figure 10). The simulation was performed on a Hartmann mask with 3 µm square holes and *d* = 8 µm pitch. The simulated plane is made of 5000 × 5000 points and each Gaussian source has standard deviation *σ* = 0.03 µm. The distance source-mask is *z*_1_ = 5 cm, the distance mask-detector is set at the first Talbot plane *z_T_* = 93 cm. The incident energy is set at 9 keV and the pixel size of the detector is 0.44 µm.

Illuminating the Hartmann mask with a fully coherent beam (Figure 10a) reproduces the periodic square apertures of the mask, as expected. Decreasing the level of coherence of the source induces a blurring effect (Figure 10b,c) as well as a superposition of the signal coming from different holes. At the highest number of individual sources (*N* = 1000, Figure 10d) the image of the periodic structure is completely lost.

Single line plots of the 2D image shown in Figure 10 are given in Figure 11. This last example shows the consistency of our simulation given a certain degree of coherence of the source: the system is reproducing the exact period structure at the Talbot distance as a consequence of the diffraction interference phenomena, while this effect cannot be seen in the case of an incoherent illumination.

The detectability of interference patterns from a test object (Section 3.1) was studied, varying the degree of spatial coherence of the source. This first result was applied in the case of a compact Hartmann sensor (Section 3.2) to study the spot shape with respect to the source properties. A standard hard X-ray Hartmann sensor (Section 3.3) was then simulated to investigate the presence of cross-talk and the spot detection. Finally, the occurrence of a classical effect from coherent illumination (Talbot effect, Section 3.4) was tested for different configurations of the source.

## 4. Conclusions

In this article we have simulated the performance of the Hartmann sensor for different degrees of coherence of the source. We first validate the Fresnel propagation method reproducing well-known diffraction patterns for a single coherent source. Next, the source was designed and tested in order to control its degree of coherence, from the completely coherent to the incoherent case. A low-density material (PMMA) was chosen to correlate the defined coherence with the presence of diffraction. Finally these findings were applied to the optimization of the Hartmann mask simulating several experimental conditions. The tool presented here is capable of predicting the diffraction patterns and the presence of cross-talk for a defined Hartmann mask. The general parameters of an experimental set-up (i.e., distances, hole size) can thus be optimized according to any source specification. The findings reported in this paper can be used for the improvement of image reconstruction and expanding the applications of X-ray phase-contrast imaging towards materials characterization and medical imaging.

## Figures and Tables

**Figure 1 sensors-20-06469-f001:**
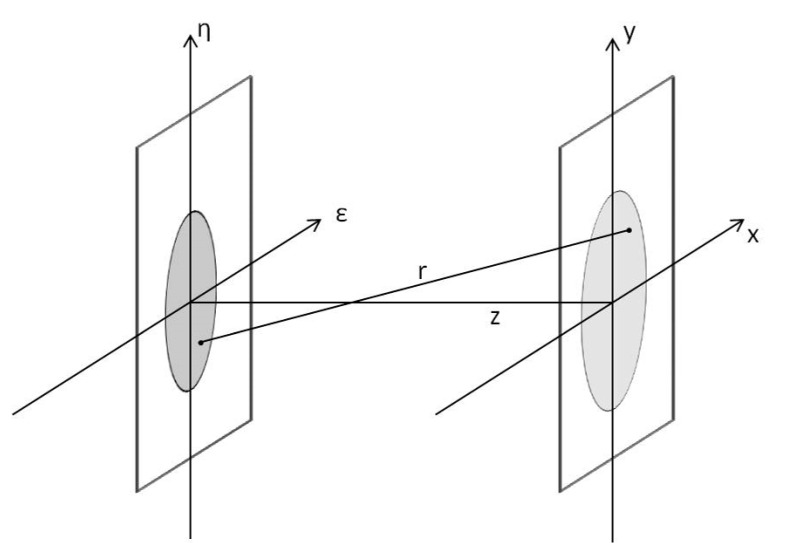
Schematic representation of the coordinates system: the wave field is defined in the (*ε*,*η*) plane and propagated until the second plane (*x*,*y*). The propagation direction z is orthogonal to both planes.

**Figure 2 sensors-20-06469-f002:**
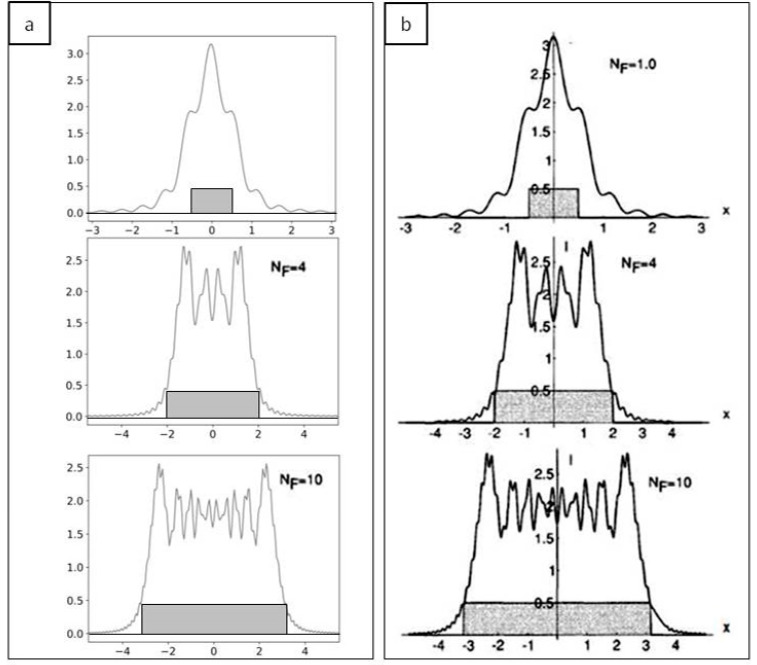
Diffraction patterns from a rectangular aperture changing the Fresnel number (*N_f_*). (**a**) Simulation results for *N_f_* = 1, 4, 10; aperture size is shown in dark grey. (**b**) Theoretical results for *N_f_* = 1, 4, 10; aperture size is shown in dark grey.

**Figure 3 sensors-20-06469-f003:**
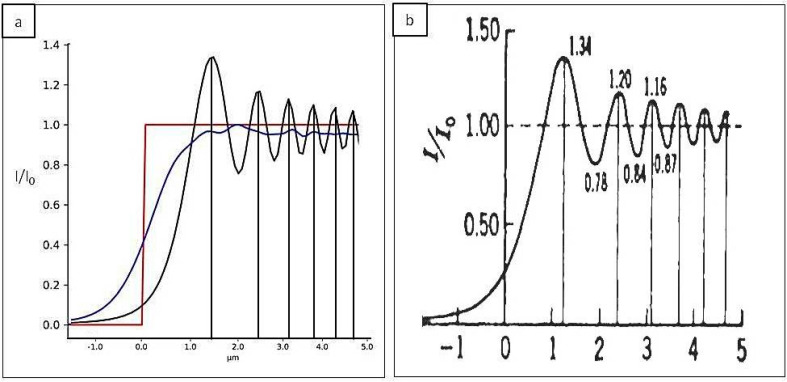
Diffraction pattern from a plane wave incident to a sharp edge in the case of coherent and incoherent illumination. (**a**) Result from the simulation for *N* = 5 (black line) and *N* = 100 (blue line). (**b**) Theoretical result for coherent illumination.

**Figure 4 sensors-20-06469-f004:**
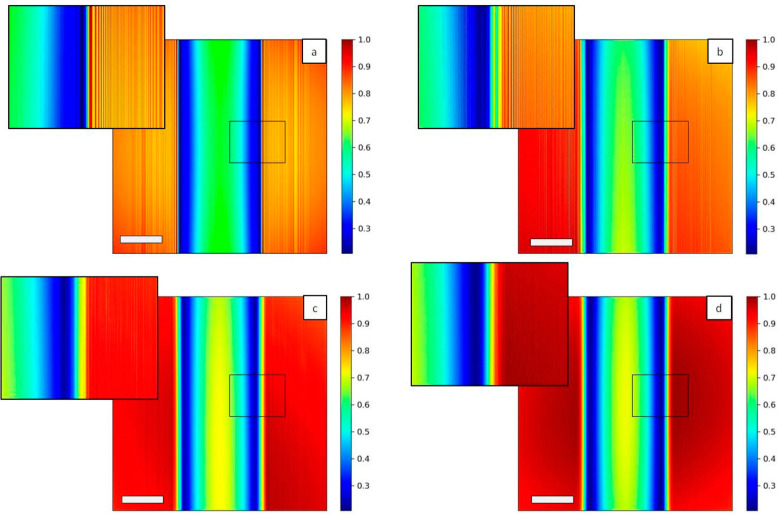
2D intensity map of a Polymethylmethacrylate (PMMA) cylinder normalized to the incident beam for different values of *N*: (**a**) *N* = 1, (**b**) *N* = 10, (**c**) *N* = 100 and (**d**) *N* = 1000. Diffraction lines can be seen in the case of full coherence (**a**). For higher *N*, the oscillations attenuate (**b**,**c**), while the image is smoothed for incoherent illumination (**d**). Scale bars are 25 µm.

**Figure 5 sensors-20-06469-f005:**
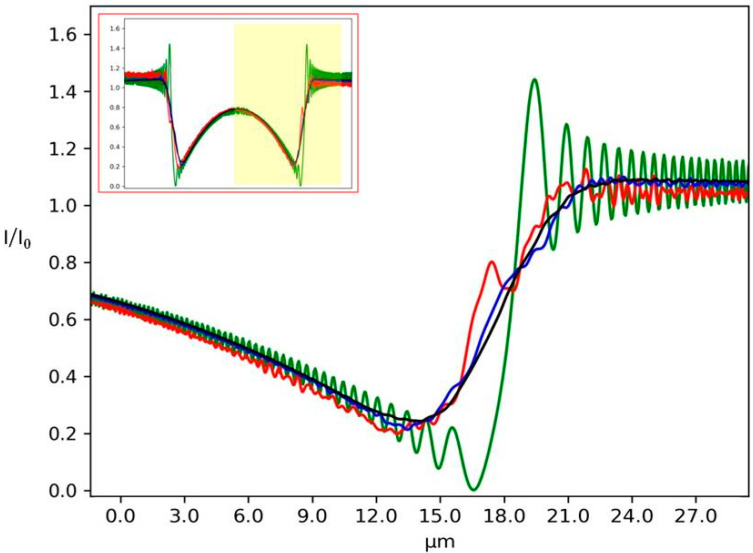
Single line plot of the 2D images displayed in Figure 4. Green line (*N* = 1) represents the case of fully coherent illumination, while the red line is for *N* = 10, the blue for *N* = 100 and the black for *N* = 1000. The inset shows the full line plot.

**Figure 6 sensors-20-06469-f006:**
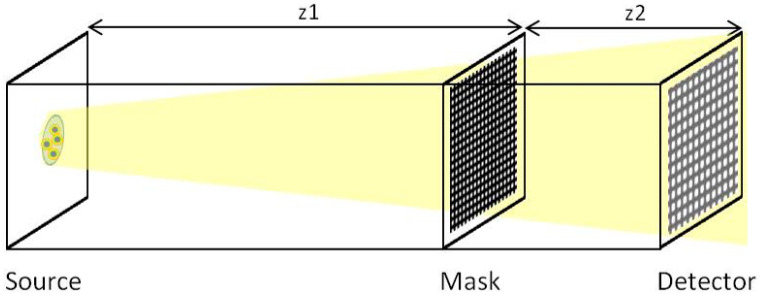
Schematic representation of the simulated set-up: the Hartmann mask is placed in the beam path between the source and the detector. *z*_1_ is the source to mask distance and *z*_2_ is the mask to detector distance.

**Figure 7 sensors-20-06469-f007:**
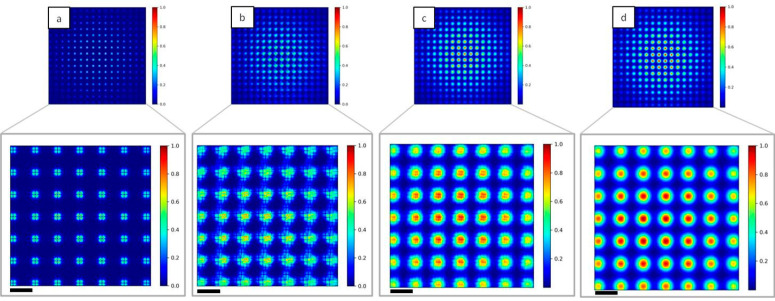
2D intensity maps of the simulated Hartmann mask imaged at the detector plane for different values of *N*. (**a**) *N* = 1, (**b**) *N* = 10, (**c**) *N* = 100 and (**d**) *N* = 1000. The diffraction pattern created with coherent illumination (**a**) will become more noisy increasing *N*, reaching a Gaussian shape in the incoherent case (**d**). Insets show a magnification of the central part of the mask. Scale bars are 16 µm.

**Figure 8 sensors-20-06469-f008:**
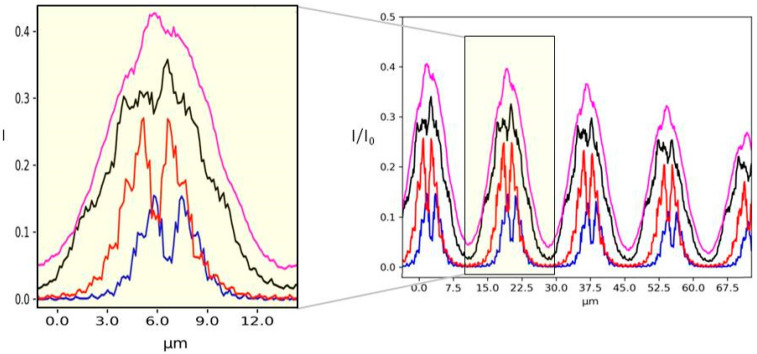
Single line plot of the 2D images shown in Figure 7 varying *N*. *N* = 1 (blue line), *N* = 10 (red line), *N* = 100 (black line) and *N* = 1000 (purple line).

**Figure 9 sensors-20-06469-f009:**
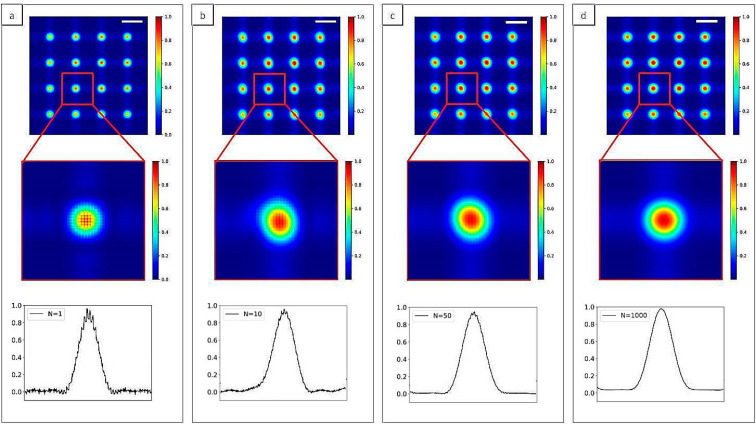
2D Intensity map of Hartmann mask with 3 µm square apertures and 25 µm pitches for different illuminating sources. The number of Gaussians N inside the source are *N* = 1 (**a**), *N* = 10 (**b**), *N* = 50 (**c**) and *N* = 1000 (**d**). Oscillations can be seen in the detected spots with coherent illumination (**a**) while they become quite circular in the incoherent case (**d**). Scale bars are 38 µm.

**Figure 10 sensors-20-06469-f010:**
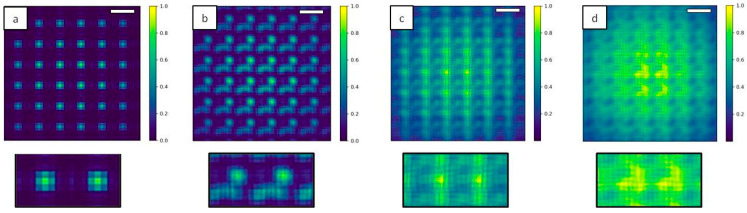
2D intensity map of the Hartmann mask at the Talbot plane increasing N. For *N* = 1 (**a**) the square pattern of the mask is exactly reproduced, while for N = 10 (**b**) and for *N* = 100 (**c**) the image is blurred. In the incoherent case *N* = 1000 (**d**) the square shape is completely lost. Insets show a magnification of the central part of the mask. Scale bars are 15 µm.

**Figure 11 sensors-20-06469-f011:**
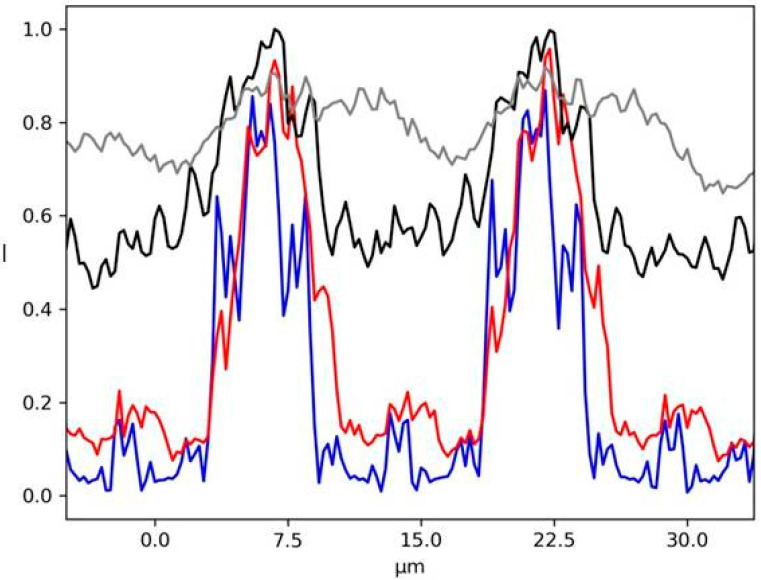
Single line plot of the images shown in Figure 9. *N* = 1 (blue line), *N* = 10 (red line), *N* = 100 (black line) and *N* = 1000 (gray line).

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
