# Peer review of "Modelling of Phase Contrast Imaging with X-ray Wavefront Sensor and Partial Coherence Beams"

_sensors, 2020, doi:10.3390/s20226469_

Round 1

Reviewer 1 Report

  1. Please, I did not see the importance of your study in the Abstract, I believe it is important to insert.
  2. Introduction (25-54). Divide / break the paragraphs, so that the work has the same number of paragraphs in its introduction.
  3. The terms of the equations are not defined, so if possible define after the insertion of the equations. Do this for all equations.
  4. To improve the quality of Figure2, the font inserted in the image is small.
  5. What is the Y axis of Figure3a?
  6. Improve the font quality of Figure9. They all seem small and difficult to understand.

Reviewer 2 Report

The paper presents a 3D wave propagation model based on Fresnel propagator to handle different ways of using an X-ray Hartmann wavefront sensor.

The topic is interesting, the paper is well written, and the innovation is evident. However, I would like to suggest some improvements.

Minor improvements:

  • figures 2b and 3b are not at the same quality as their respective counterparts.
  • The literature review does not represent the state of the art from the field. Just a few references newer than 2015.

Major improvements:

  • It is a question of style, but I have missed a general discussion in section 3. The authors have presented subsections 3.1, 3.2, and 3.3 individually, but there was no closing in this section, no general comment comparing all the approaches together. 
  • Finally, the only critical point I see in this work is the lake of comparison with other approaches to comprove the effectiveness of your methodology. 

Reviewer 3 Report

In general, this work is interesting and contains valuable knowledge. it presents the results of the development and testing of the model of the Hartmann sensor for different degrees of coherence of the source. with the exception of small comments presented below, I consider the work positive and admitting it to publication.

Comments:

1. There are inaccuracies in the formulas, for example: in formula (2.1) the limits of integration are not indicated; (2.4) and (2.6) are not readable, in (2.7) bracket is missed before the multiplication sign, since otherwise, the expression is meaningless, since the direct and inverse Fourier transforms of the multiplication of two functions is performed.

2. In this work, a mathematical experiment is carried out, however, the feasibility of X-ray detectors with a pixel size of hundreds of nanometers is not entirely clear.

3. In my opinion, the mathematical model developed by the authors is poorly described, it is not noted in what the uniqueness of the approaches used by the authors and key differences from the existing ones 

Round 2

Reviewer 2 Report

all my questions were responded.